# Machine Learning-Based Prediction of Drug-Drug Interactions for Histamine Antagonist Using Hybrid Chemical Features

**DOI:** 10.3390/cells10113092

**Published:** 2021-11-09

**Authors:** Luong Huu Dang, Nguyen Tan Dung, Ly Xuan Quang, Le Quang Hung, Ngoc Hoang Le, Nhi Thao Ngoc Le, Nguyen Thi Diem, Nguyen Thi Thuy Nga, Shih-Han Hung, Nguyen Quoc Khanh Le

**Affiliations:** 1Department of Otolaryngology, Faculty of Medicine, University of Medicine and Pharmacy at Ho Chi Minh City, Ho Chi Minh City 70000, Vietnam; luonghuudang167@gmail.com (L.H.D.); quang.lx@umc.edu.vn (L.X.Q.); 2Department of Rehabilitation, Da Nang Hospital of C, Da Nang City 50000, Vietnam; bsnguyentandungbvc@gmail.com; 3Department of Rehabilitation, Da Nang University of Medical Technology and Pharmacy, Da Nang City 50000, Vietnam; 4Department of Otolaryngology, University Medical Center, Ho Chi Minh City 70000, Vietnam; lequanghungdr9999@gmail.com; 5Graduate Institute of Biomedical Materials and Tissue Engineering, College of Biomedical Engineering, Taipei Medical University, Taipei City 110, Taiwan; lengochoang252@gmail.com (N.H.L.); lethaongocnhi@gmail.com (N.T.N.L.); 6Department of Otolaryngology, Cai Lay Regional General Hospital, Cai Lay 84000, Vietnam; bsnguyenthidiem@gmail.com; 7Faculty of Nursing and Midwifery, Hanoi Medical University, Ha Noi 10000, Vietnam; nguyenthuynga@hmu.edu.vn; 8International Master/Ph.D. Program in Medicine, College of Medicine, Taipei Medical University, Taipei City 110, Taiwan; 9Department of Otolaryngology, School of Medicine, College of Medicine, Taipei Medical University, Taipei City 110, Taiwan; 10Department of Otolaryngology, Wan-Fang Hospital, Taipei Medical University, Taipei City 106, Taiwan; 11Professional Master Program in Artificial Intelligence in Medicine, College of Medicine, Taipei Medical University, Taipei 106, Taiwan; 12Research Center for Artificial Intelligence in Medicine, Taipei Medical University, Taipei 106, Taiwan; 13Translational Imaging Research Center, Taipei Medical University Hospital, Taipei 110, Taiwan

**Keywords:** drug-drug interaction, histamine antagonist, machine learning, PyBioMed package, cheminformatics, SMILES

## Abstract

The requesting of detailed information on new drugs including drug-drug interactions or targets is often unavailable and resource-intensive in assessing adverse drug events. To shorten the common evaluation process of drug-drug interactions, we present a machine learning framework-HAINI to predict DDI types for histamine antagonist drugs using simplified molecular-input line-entry systems (SMILES) combined with interaction features based on CYP450 group as inputs. The data used in our research consisted of approved drugs of histamine antagonists that are connected to 26,344 DDI pairs from the DrugBank database. Various classification algorithms such as Naive Bayes, Decision Tree, Random Forest, Logistic Regression, and XGBoost were used with 5-fold cross-validation to approach a large-scale DDIs prediction among histamine antagonist drugs. The prediction performance shows that our model outperformed previously published works on DDI prediction with the best precision of 0.788, a recall of 0.921, and an F1-score of 0.838 among 19 given DDIs types. An important finding of the study is that our prediction is based solely on the SMILES and CYP450 and thus can be applied at the early stage of drug development.

## 1. Introduction

Many factors influence drug metabolism, of which the majority of drug metabolism is due to isozymes [1]. Numerous statistics have shown that more than 80% of drug-based drug metabolism is related to the cytochrome P450 family [1,2]. In most cases, drug metabolism in the human body is mediated by an enzyme belonging to the CYP450 group or by many different enzymes. The effects of drugs on the human body or the interactions between drugs and enzymes are divided into two main categories: induction and inhibition. In 1997, B A Sproule et al. investigated the inhibitory capacity of Sertraline (Zoloft) against the CYP2D6 enzyme at different doses. They showed that inhibition became more prominent when the dose was increased to 200 mg instead of 50 mg. Moreover, the combination of drugs with the same metabolic enzyme can also lead to undesirable side effects by inhibiting/inducing mechanisms. For instance, based on the report of Tom Lynch, the inhibitory enzyme of Metronidazole, CYP2C9, is also the metabolic enzyme of Warfarin [2]. Concomitant administration of Metronidazole and Warfarin leads to an increase in the blood concentration of Warfarin due to CYP2C9 inhibition by Metronidazole but also leads to an increased risk of bleeding [2]. Every year, the US FDA announces numerous withdrawals of drugs from the market or makes recommendations for their use due to adverse effects [3]. The nonsedating antihistamines terfenadine (Seldane) and astemizole (Hismanal) and the gastrointestinal peristalsis cisapride (Propulsid) were all withdrawn from the US market due to their ability to inhibit the metabolism of other drugs, leading to arrhythmia and a life-threatening condition for the patient [4]. The calcium channel blocker, mibefradil (Posicor), was also withdrawn from the US market in 1998 because it is a potent enzyme inhibitor that contributes to the toxicity of other cardiovascular drugs.

Adverse drug effects are more common when using drugs that belong to the “blockbuster” drug class (very popular drug), as determined by their popularity. As reported by Lotte Berghauser Pont et al. in 2018, more than half of the sales of the top 20 pharmaceutical companies are dependent on blockbuster drugs, and the trend to focus on developing these drugs will continue to increase. Despite the emphasis on pharmacovigilance, blockbuster drugs can still be withdrawn following events that occur long after their release. Rofecoxib (Vioxx) was one of the most anticipated blockbuster drugs in 1999. Rofecoxib was a novel anti-inflammatory cyclooxygenase-2 (COX-2) inhibitor and promised to contribute to improving pain for millions of patients worldwide, but it was withdrawn in 2004 due to an increased risk of cardiovascular events [3,5]. Histamine antagonists were among the first ‘blockbuster’ drugs, with a focus on treatments for allergy, anxiety, and insomnia. Typically, people use histamine antagonists as inexpensive, generic, nonprescription drugs that can provide relief from the common cold or allergy symptoms (nasal congestion, sneezing, hives, or animal allergies) with few side effects [6,7,8]. Histamine antagonists, such as hydroxyzine, have been used as successful treatments in generalized anxiety disorder (GAD) and are often used in posttraumatic stress disorder (PTSD) and acute anxiety [9,10,11]. Some histamine antagonists are often prescribed as sedative treatments and have to use it frequently. It is dangerous if the patient is using histamine antagonists concurrently with other drugs without specific instructions from a physician, as they may cause adverse interactions in the nervous system [12,13,14].

Recognizing the importance of a pharmacovigilance assessment for histamine antagonist drugs and related drugs, in this study, we provided a machine learning-based model using the simplified molecular-input line-entry system (SMILES) of drug structure, which may be a promising tool to predict multiple drug interaction types mainly involving 73 approved histamine-antagonist drugs. SMILES (Simplified Molecular Input Line Entry System) is a notation method to simplify the elements in chemical structure of a molecule into chemical symbols [15]. At the same time, the computer can easily read and analyze SMILES symbols and thus provide good parameters for machine learning models such as QSAR [16,17] or QSPR [17]. For assessing a particular drug-drug interaction, it should be evaluated on a variety of biological factors as well as occupy a lot of time, cost and manpower to bring a new drug to market. With the use of SMILES, the drug interactions can be assessed quickly through some classification models in the computer [18,19,20], offering many advantages in the early stages of drug development compared to traditional procedure. In the past decade, many methods have been developed to predict or investigate drug-drug interactions [21,22,23]. However, due to the lack of human resources, large comprehensive medical databases or other experimental factors (time, animal, cost, etc.), computational tools/software as well as many medical-related databases, such as DrugBank [24], ZINC [25], DailyMed, and Human Metabolome Database [26], have been developed to assist in adverse drug effect evaluations. Recently, many studies and comprehensive reviews have been published addressing attempts to use machine learning or other computational methodologies in the evaluation of drug-drug interactions [20,27,28]. For example, Assaf Gottlieb et al. gathered drug interactions, indications, and side effects data over 50 years from 1961 to 2010 to reveal the correlation between drug indications and their pharmacodynamics. They discovered a new method, “Inferring Drug Interactions”, which can allow people to evaluate the common metabolizing enzymes and infer the pharmacokinetics of drugs [29]. J. You et al. combined two drug-target interactions (DTIs) datasets under protein identifiers (Version 5.0.10) and external target drug-UniProt links (Version 5.0.10) with a total of 17,331 drug-target interactions [30]. In their research, the drug feature extractions only focused on small molecule drugs; biotech drugs were filtered and removed by the Rcpi R package. Then, 14,792 known DTIs from DrugBank were classified using the SLG and LASSO regression models. The LASSO-based regularized linear classification models yielded better results than the SLG models for predicting DTIs, with accuracies and AUCs of up to 81% and 0.89, respectively. Konstantinos Bougiatiotis et al. [31] utilized topological data, which was generated by graph neural network, as input for DDI prediction. In the study, beside experimental data, topology of the structure also provided useful information and increased the performance of deep learning prediction mode. Although a plethora of statistical methods have been developed to predict DDIs based on the use of the logistic regression model to predict future interactions, most of the recent approaches still focus on three main kinds of categorizations of DDIs prediction: similarity-based, classification-based, and text mining approaches.

Although previous methods achieved great advances, more detailed predictions are still needed. We proposed a histamine antagonist interaction-network inference (HAINI) framework (Figure 1) that was applied to the SMILES and metabolism profile of 26,344 histamine antagonist drug-pairs collected from the DrugBank database; each drug-pair was independently labeled into 19 classes corresponding to their interactions. Next, we calculated drug-drug interactions in a similar manner to the features by using the PyBioMed package as a chemical-chemical interaction descriptor to extract 3600 features using the PyInteraction module [32]. Additionally, we also establish new features based on the interaction of drug pairs on the CYP450 group to combine with chemical interaction features. HAINI target models were run on five machine learning algorithms to serve as predictive models: Naive Bayes (NB), Decision Tree (DT), Random Forest (RF), Logistic Regression (LR), and XGBoost (XGB). The results were evaluated with fivefold cross-validation to approach a large-scale novel DDI. We also validated the highest prediction algorithms using another dataset collected independently from the DrugBank database, which is mainly composed of non-approved drugs. In summary, we constructed a machine learning model to collect the results of predictive models and made it available through an open access repository together with all the datasets and the results of the study at https://github.com/tair-group/HAINI accessed on 8 November 2021.

## 2. Materials and Methods

### 2.1. Data Preparation

All drug data in this study including SMILES, drug target and CYP450 metabolism profile were collected from Drugbank database [24] (version 5.1.7, released 2 July 2020). The dataset was divided into 2 groups which were approved and non-approved drug, wherein approved drugs were used as training input data and the other group was used for validation. Although the non-approved drugs were not official on the market, the interaction data of these drugs were still available and suitable for DDI prediction. We used specific descriptions of the interaction between two drugs as labels for the DDIs type, for example: “DRUG_A can cause a decrease in the absorption of DRUG_B, resulting in a reduced serum concentration and potentially a decrease in efficacy”. There was a total of 19 types of interactions based on this specific description, which were categorized from 1 to 19.

### 2.2. Data Preprocessing

The preprocessing procedure comprised 3 steps: chemical interaction feature extraction, CYP450-related interaction feature extraction and data hybridization.

#### 2.2.1. Chemical Interaction Feature Extraction

PyBioMed is a package written in the Python programming language that can be used to create numerous future vectors from molecular structure, protein sequences, and DNA sequences. PyBioMed is a remarkable tool and can be applied to a wide range of tasks in areas related to cheminformatics, bioinformatics, and systems biology [32]. The PyBioMed package includes six main modules, PyInteraction, PyDNA, PyMolecule, PyProtein, PyGetMol, and PyPretreat, to compute various molecular descriptors as well as assist in processing the input data. In this study, we mainly focused on identifying drug interactions based on chemical structure; hence, the PyInteraction module was used to calculate the features between drug pairs with interactions based on the SMILES structures of 26,344 DDI pairs. In the PyInteraction module, we used the chemical-chemical interaction descriptor to extract features of two drugs in a drug pair and multiplied them together with the following equation.
*F* = {(*k*) = *F*(*i*) × *Fb*(*j*), *i* = 1, 2, …, *p*, *j* = 1, 2, …, *p*, *k* = (*i* − 1) × *p* + *j*},(1)

This equation gave us the most features (up to 3600 features); the other two calculation functions only generated 120 features per pair of interactions.
*Fab* = (*Fa*, *Fb*),(2)
*F* = [*Fa*(*i*) + *Fb*(*i*)), *Fa*(*i*) × *Fb*(*i*)],(3)

#### 2.2.2. CYP450-Related Interaction Feature Extraction

Most of the drugs are metabolized at liver by CYP450. A majority part of DDI occurs via CYP 450 activity. For example, Drug A is a substrate of CYP3A4, which means that CYP3A4 metabolizes drug A and decrease drug A concentration. If Drug B inhibits CYP3A4, the metabolism rate of drug A declines and subsequently this leads to an increase in drug A concentration. In this model, we proposed a model to generate a CYP-related interaction feature. Particularly, the metabolism profile of each drug was summarized and encoded into vector. There were 12 CYP proteins being investigated including 1A2, 2A6, 2B6, 2C18, 2C19, 2C8, 2C9, 2D6, 2E1, 3A4, 3A5, 3A7. The interaction type was described as Drug A_Drug B in which Drug A is affected by Drug B. Drug_A and Drug_B vector was encoded for all CYP proteins and the CYP protein order was kept instantly throughout the whole dataset vector generation. For Drug_A vector, if drug A is a substrate of one certain CYP, the value of component at that CYP position is 1, otherwise the value is 0. For Drug_B vector, if drug B inhibits a certain CYP, the value at that CYP position is 1. If drug B induces the CYP, the value is −1. If there is no interaction between drug B and CYP, the value is 0. Finally, the interaction factor was generated by multiplying Vector_A_ and Vector_B_ as the equation:Interaction vector_A_B_ = Vector_A_ × Vector_B_(4)

Detailed information for feature extraction from CYP450 groups has been shown in Table 1.

### 2.3. Development of the Predictive HAINI Model

In this study, the drug interaction classification model (HAINI) was built based on the combination of two types of features: (1). Chemical properties extracted from SMILES of interacting drug pairs; (2). Features are generated from the calculation from the interaction vector of the CYP450 group of drug pairs.

The model training process is performed based on 70% of the input data with a combination of hyperparameter tuning (Table 2), cross-validation (k = 5). The remaining 30% of the data is used to test the model efficiency. In addition to being able to somewhat confirm the predictability of drug interactions, we used another dataset that included drugs of the “investigational, experimental” group that interact with histamine antagonist drugs (Appendix A).

Imbalanced data class has been causing trouble for predictive classification in machine learning. Regularly, the minority class was ignored in the model. However, it was seen in some cases that these minority classes gave the most important performance. There are 2 common ways which reduce the imbalance of the dataset including oversampling the minority class or undersampling the majority class. In this model, Synthetic Minority Oversampling Technique (SMOTE) was applied as a data augmentation tool. This technique simply duplicates data in the minority class to balance the dataset while not adding new information.

We applied a total of five machine learning algorithms in the classification process including: Naive Bayes (NB), Decision Tree (DT), Random Forest (RF), Logistic Regression (LR), and XGBoost (XGB).

▪Naive Bayes (NB)

In this study, the Naive Bayes method was applied as a supervised learning algorithm for our dataset [33]. The posterior probability of our dataset was determined using the equation below:(5)$$P\left({c|x}_{1}{,\ldots ,\;x}_{n}\right)=\frac{P\left(c\right){\times P(x}_{1}{,\ldots ,\;x}_{n}\left|c\right)}{{P(x}_{1}{,\ldots ,\;x}_{n})}$$
-$P\left({c|x}_{1},\ldots {,\;x}_{n}\right)$ is the posterior probability of a class (c, target) of a given predictor (x, attributes).-$P\left(c\right)$ is the prior probability of a class.-${P(x}_{1}{,\ldots ,\;x}_{n}\left|c\right)$ is the likelihood, which is the probability of a predictor of a given class.-${P(x}_{1}{,\ldots ,\;x}_{n})$ is the prior probability of the predictor.-Vector ${(x}_{1}{,\ldots ,\;x}_{n})$ represents some n features.
▪Decision Tree (DT)

In the field of machine learning, Decision Tree (DT) is a predictive model, that is, it generates a map from observations of an object/phenomenon to allow conclusions to be drawn about the target value of the object/phenomena [34]. In this study, the data were given as records of the form:(X, Y) = (X_1_, X_2_, X_3_ …, X_i_, Y),(6)

In which the interaction types or Y are denoted (19 classes-dependent variables) as the variables for prediction. X_1_, X_2_, X_3_, and so on are variables equivalent to descriptors that act as input data to contribute to the decision of the type of interaction between the given drug pair.

▪Random Forest (RF)

Random forest (RF) or Random Decision Forest describes an ensemble learning algorithm that was first introduced in 1995 [35]. This statistical learning methodology is used for classification, regression, and other tasks that operate by generating multiple decision trees at training time and outputting the class that is the mode of the classes (classification) or the mean prediction (regression) of the individual trees in the forest implemented on three similarities of the DDIs.

▪Logistic Regression (LR)

The Logistic Regression (LR) method is a regression model that predicts a discrete target variable Y (DDI types) corresponding to an input vector X (106 features) [36]. This is equivalent to whether the feature (X) extracted from the preprocessed dataset belongs to any of the 19 classes.

▪XGBoost (XGB)

XGBoost is an optimized software library whose algorithms derive from gradient boosting framework [37]. Gradient boosting framework belongs to tree-based model for regression, classification and prediction beside decision tree and random forest. Gradient boosting is known as an ensemble of many trees. The ability to boost a “weak” model into a “stronger” model by optimizing an arbitrary differentiable loss function makes it outstanding from the other 2 tree-based model.

## 3. Results

### 3.1. Evaluation of HAINI Performance

In this study, we trained and tested the HAINI model on the compiled drug-drug interaction (DDI) dataset from the DrugBank database [24] (version 5.1.7, released 2 July 2020). Here, we focused on adverse drug interactions where the chosen drugs belong to the histamine antagonist group or interact with them.

The framework of HAINI, which is clearly described in Figure 1, consists of the following three main steps: (i) filtering drugs (including drug ID, SMILES, and detailed interaction) of the histamine antagonist group and drugs that interact with them from the DrugBank database; (ii) labeling the interaction types and extracting features of the interactive drug pairs based on CYP450 (Appendix A), SMILES and the PyInteraction module; and (iii) applying various common classifier algorithms, such as Naive Bayes, Decision Tree, Random Forest, Logistic Regression, and XGBoost. All machine learning models have been implemented and visualized using Python scikit-learn and matplotlib packages.

The training process was performed based on 70% of the data from the pre-processed dataset, and the remaining 30% is used as testing data. We used the cross-validation technique (k = 5) for the training of the HAINI model. Of these, more than 18,000 histamine antagonist drug pairs (70% of pre-processed data) were randomly divided into five equal-sized subsets to perform the training process. In order to evaluate the model performance, several metrics were applied for all learning algorithms, i.e., Precision, Recall, F-measure (F1) as follows [38,39]:Precision = TP/(TP + FP)(7)
Recall = TP/(TP + FN)(8)
*F*–1 score = (2 × Precision × Recall)/(Precision + Recall)(9)
Accuracy = (TP + TN)/(TP + TN + FP + FN)(10)
where TP, FP, TN, and FN stand for True Positive, False Positive, True Negative, and False Negative, respectively. Precision is the ratio between the numbers of correctly predicted DDIs and all DDIs; recall is the ratio between number of correctly predicted DDIs and all true DDIs. Since recall and precision affect each other, improving one of them may lead to a reduction in the other. Therefore, we used the *F*-1 score, which is the geometric mean of precision and recall, to make the predictive results more reasonable.

### 3.2. Improvement of HAINI Performance

Based on the results of the HAINI model for raw data, we easily observed that XG-Boost was the best performing algorithm compared to the other four algorithms. Even with the highest precisions of some classes being greater than 80%, we could also clearly observe an imbalance of this dataset according to the value of Recall (the difference between recall and precision is usually greater than 40%). In the pre-processed dataset, only 7 out of 19 classes had a number of drug pairs exceeding 1000 drug pairs (as shown in Figure 2). This finding clearly shows imbalances in the input data; therefore, we made a number of improvements in the model to improve the recall and precision values evenly.

#### 3.2.1. Feature Selection

To improve the gap between the precision value and the recall value, we implemented the selection of important features by using the Random Forest and BestFit ranking methods on pre-processed data. From a total of 3600 features extracted from the PyInteraction module, we filtered out the 94 most important features (in both ranking methods) in classifying the interaction types.

Based on the ranking of the top 20 most important features represented by a Shap Summary Plot (Figure 3), we can clearly see that the distribution of the top feature ranked by the BestFit method (the most related to features MRVSA2 and MRVSA6) were more centralized than in the Random Forest method. At the same time, the ranking of the important features by the BestFit method also presents higher feature values than the Random Forest method. Therefore, the HAINI model’s accuracy also increased from 0.59 to 0.63 when using only chemical features ranked by the BestFit method.

#### 3.2.2. Applying Synthetic Minority Oversampling Technique (SMOTE)

Although the accuracy of the HAINI model underwent a significant increase (4%) by using the BestFit ranking method, some interaction types still have large discrepancies in accuracy values due to the small amount of input data. Therefore, we constructed a simple approach to enhance the imbalanced interaction type by augmenting the minority classes with the SMOTE.

The 94 features extracted from SMILES were combined with features extracted from the interaction vector of the CYP450 group (Appendix A), used as training data. The model’s results are significantly improved when the model’s accuracy reaches 0.921 with the XGBoost classifier, a marked increase compared to using only features from SMILES (78% of accuracy). The ROC curve in Figure 4 shows the classification performance of the five classifiers mentioned above. Therein, Decision Tree gives the highest curve result with AUC of 0.966. However, in the results of each class, we found that this classifier shows signs of overfit when it gives abnormally high results (AUC is higher than 0.9 in all classes). We therefore believe that the results of the XGBoost classifier will be more reliable (Table 3).

### 3.3. HAINI Performance on the Validation Dataset

To further demonstrate the reliability and robustness of HAINI, we also assessed its performance on another dataset that was compiled from the DrugBank database, which also contains only 19,971 DDI pairs from the DrugBank database with the status “experimental” and “investigational”. The top five predicted results of XGBoost classifier are presented in Table 4.

The results of the validation show that the highest precision with the highest precision belongs to interaction type No. 3, with a precision value of 0.73. At the same time, most of the DDI types having the highest precision values of the validation dataset are the same as the top DDI types of the training dataset. For example, in all the top five classes of the validation dataset, the predictive results were achieved without any significant difference compared to the result of the training process (Table 3). Specifically, ziprasidone (DB00246) is a drug used for the treatment of schizophrenia and also has a low affinity to histamine H1 receptors. We successfully tested the HAINI model on the validation dataset and correctly predicted the interaction between ziprasidone (DB00246) and Idazoxan (DB12551) compared to the authenticated DrugBank database.

Although there were some DDI classes with relatively low predictive results, considering the amount of input data, we observe that the number of drug pairs used in the prediction is many times less than the DDIs with high precision.

### 3.4. Comparison of HAINI Performance on Previous Studies Using Chemical Similarity

Compared with previous studies [40,41,42], using the PyInteraction module (from PyBioMed package) to extract features from the chemical structures of drug pairs, the idea of combining features from CYP450 groups and features from SMILES significantly improved the working performance of the HAINI machine learning model (Table 4). Compared to a binary prediction (with or without adverse drug interactions), a predictive model of multi-adverse interaction types with an average precision of approximately 0.783 is acceptable. In addition, for each kind of interaction, the HAINI predictive model has different efficiencies, in which the best efficiency of the model with each separate class has a precision value of up to 0.921. The results confirmed that the essentials of chemical structure similarity yielded significantly discriminating features in predicting DDIs instead of only focusing on the integration of various similarities. For reference studies, most of the authors used chemical structure data from the DrugBank database, which ensures fairness in comparisons between results in using chemical structures to classify drug interactions. It can be said that the use of histamine antagonist-related drug interactions is a subset of drug interaction studies using chemical structure (SMILES). When compared with Yifan Deng’s research [18], we can see a modern method with high accuracy when using a combination of many “Omics” (substructure, target, enzyme, pathway) to predict 65 type of interactions from 74,528 pairwise DDIs.

## 4. Discussion

In this paper, our main goal was to use the properties of the chemical structures (SMILES) of interactive drug pairs to generate accurate predictive models in combination with CYP450-related interaction features. The hybrid features will save time in having to use encoders to simplify input data, avoid overfitting in the machine learning model, and help to facilitate the accurate prediction of drug interactions as well as adverse drug interactions.

In addition, while our model has a simple input type, many features (approximately 3600) can still be produced with each pair of drugs when using the PyInteraction module, and the predicted value of the HAINI model in each class (equivalent to a detailed interaction type) is also very high (over 0.921). With a large amount of attention drawn to drug-drug interactions in the field of computer science research, computational predictions using machine learning are highly encouraged. Compared to the results of some previous studies using only chemical similarity in machine learning or deep learning models (Table 4), the direct use of interactive features generated from interactions of drug pairs has a stronger effect than when separating the features of each structure and then building the matrix network or assessing the similarity of each drug with each other. For example, compared with the Deep Learning-Neural Network model of Narjes Rohani and colleagues [41], we can see that the creation of a matrix comparing the similarity between SMILES structures of drug pairs serves as a diagnostic database for precision, which is significantly lower than in the HAINI model (Table 5), while HAINI is also a multiclass model and is clinically more significant than a binary model (allowing only cases for which an interaction is present).

It is essential to have a model to accurately predict DDIs for the reasons outlined above, such as saving on testing costs and quickly identifying drug interactions that cause many side effects. Such models are also effective tools to support doctors in prescribing drugs to people who are taking different medications at the same time. When prescribing a drug to a patient, it is more helpful to specifically identify the details of the interaction between drugs than only determining whether the drug has side effects. Some drugs sometimes have only mild side effects for patients that are within acceptable levels; therefore, only defining a binary class problem with or without information about an interaction does not make sense in contributing to clinical research or drug development. Moreover, in our study, we found that the high predictability rates of some types of interactions are not entirely dependent on the amount of data input (i.e., number of interacting drug pairs). We can clearly observe this based on Appendix A in interaction types 15, 6, and 1; the number of input data pairs of these interactions is much lower than the top classes shown in Figure 2. In addition, most of the chemical subclasses of drugs that interact with histamine antagonists are often very diverse, so it is difficult to confirm whether the prediction rate depends on chemical classifications.

Currently, although the quality of health management in the world is increasing, the amount of histamine antagonists prescribed, as well as the cost of treating related diseases, have not shown signs of decline. The management and evaluation of factors related to using antihistamine or histamine antagonists in clinical treatments are necessary to improve the quality of medical treatments and to reduce the economic burden on the health care system. This study shows the potential application of computer science technology to reduce unnecessary costs of evaluating clinical drug interactions, especially with the incorporation of a machine learning method. Using machine learning analysis could lead to the identification of many potential undiscovered interactions and will also reduce the cost of clinical trials of new drugs. Moreover, there will be more potential in the analysis of drug interactions when various genomic, chemical, or other datasets related to human metabolism processes are combined. The limitation in our analysis is that the findings are only based on large-scale databases and do not include more relevant clinical databases (e.g., Drugs.com, Medscape Multi-Drug Interaction Checker, RxList). Nevertheless, through this study, we have shown the potential use of these downloadable databases, which would greatly expand the possibilities of massive data mining in the medical field.

In the future works, features will be ranked to prioritize those with the potential to influence types of drug pair interactions. We will also apply new deep learning models to increase the ability to accurately predict drug interactions and to detect unpredicted interactions.

## 5. Conclusions

In this paper, we proposed a new CYP450-related interaction feature extraction model and a new multilayer prediction model, HAINI, using machine learning in conjunction with the PyBioMed package to extract DDIs. The performance of the HAINI model is robust based on a single similarity and is ready to integrate more drugs and target-related information. Besides, the addition of CYP450-related interaction features significantly improved the performance of the model when compared to using SMILES as the only input data. The results of the tests on both the training and validation datasets are high for most specific types of interactions. In the future, we will continue to explore other structures of the neural network for multitasking learning to improve the performance of the model to detect DDIs of various drugs at the same time, not only for drugs of the histamine antagonist group.

## Figures and Tables

**Figure 1 cells-10-03092-f001:**
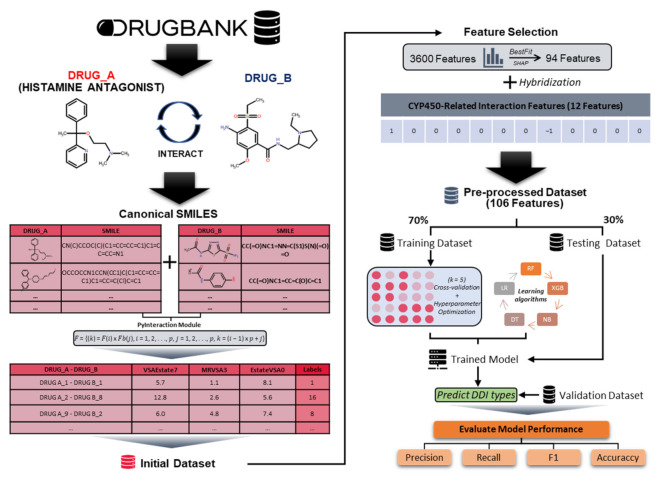
The architecture of the heterogeneous histamine antagonist interaction-network inference (HAINI) framework for predicting multiclass drug-drug interactions (DDI).

**Figure 2 cells-10-03092-f002:**
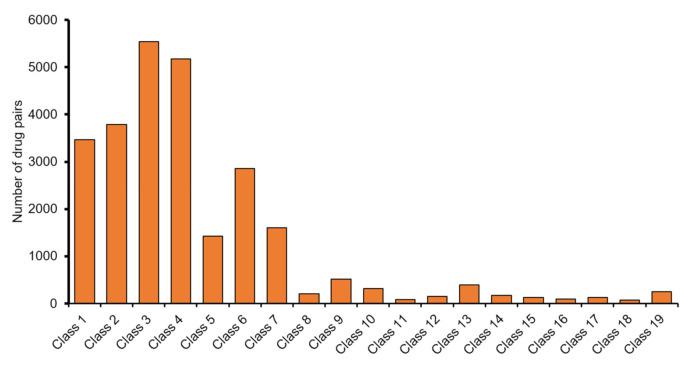
The number of drug pairs of 19 DDI types (classes) after cut-off low data classes.

**Figure 3 cells-10-03092-f003:**
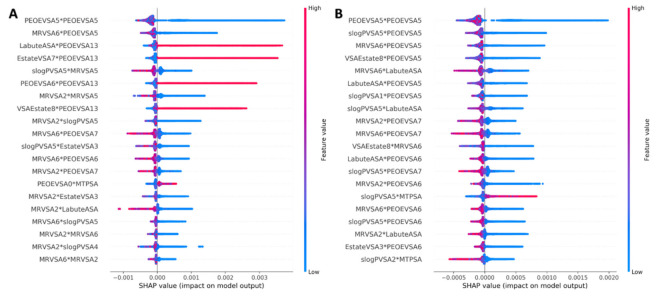
Shap summary plot of top 20 important features ranked by BestFit method (**A**) and Random Forest method (**B**).

**Figure 4 cells-10-03092-f004:**
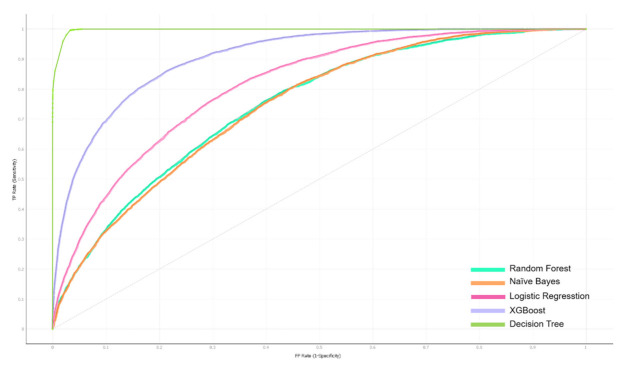
Receiver Operating Characteristic (ROC) curves of five machine learning classifiers.

**Table 1 cells-10-03092-t001:** Description of the feature extraction from CYP450 groups.

Vector_A_					Vector_B_				
1A2	Subtrate	1	Vector_A_ =	1	1A2	No interaction	0	Vector_B_ =	0
2A6	Non-subtrate	0		0	2A6	No interaction	0		0
2B6	Non-subtrate	0		0	2B6	Inhibitor	1		1
2C18	Non-subtrate	0		0	2C18	No interaction	0		0
2C19	Non-subtrate	0		0	2C19	No interaction	0		0
2C8	Non-subtrate	0		0	2C8	Inhibitor	1		1
2C9	Non-subtrate	0		0	2C9	No interaction	0		0
2D6	Non-subtrate	0		0	2D6	Inducer	−1		−1
2E1	Non-subtrate	0		0	2E1	No interaction	0		0
3A4	Subtrate	1		1	3A4	Inducer	−1		−1
3A5	Non-subtrate	0		0	3A5	No interaction	0		0
3A7	Non-subtrate	0		0	3A7	No interaction	0		0

**Table 2 cells-10-03092-t002:** Hyperparameter search grid and optimal value for each machine learning algorithm.

Algorithms	Hyperparameter Grid		Optimal Parameter
Naïve Bayes	C:	0.001, 0.01, 0.1, 1, 10, 100, 1000	C: 100
	gamma:	0.1, 0.2, 0.3, 0.4, 0.5, 0.6, 0.7, 0.8, 0.9, 1.0	gamma: 0.5
	kernel:	rbf, linear	kernel: rbf
Logistic Regression	Penalty:	11, 12	Penalty: 12
	C:	0.001, 0.01, 0.1, 1, 10, 100, 1000	C: 100
Decision Tree	criterion:	gini, entropy	criterion: Gini
	max_depth:	10, 20, 30, 40, 50, None	max_depth: 10
	min_samples_leaf:	1, 2, 4	min_samples_leaf: 2
	min_samples_split:	2, 5, 10	min_samples_split: 5
Random Forest	bootstrap:	True, False	bootstrap: False
	max_depth:	10, 20, 30, 40, 50, None	max_depth: 10
	max_features:	auto, sqrt	max_features: sqrt
	min_samples_leaf:	1, 2, 4	min_samples_leaf: 2
	min_samples_split:	2, 5, 10	min_samples_split: 5
	n_estimators:	20, 40, 60, 80, 100, 200, 500, 1000, 1500	n_estimators: 1500
XGBoost	max_depth:	10, 20, 30, 40, 50, None	max_depth: 10
	max_features:	auto, sqrt	max_features: sqrt
	min_samples_leaf:	1, 2, 4	min_samples_leaf: 2
	min_samples_split:	2, 5, 10	min_samples_split: 5
	n_estimators:	20, 40, 60, 80, 100, 200, 500, 1000, 1500	n_estimators: 1500

**Table 3 cells-10-03092-t003:** Top 10 DDI types having highest precision when classifying with XGBoost algorithm (5-fold cross-validation).

Type of Interactions *	Recall	Precision	F1-Score	Rank
Class 13	0.906	0.904	0.999	1
Class 15	0.839	0.838	1.000	2
Class 6	0.837	0.818	0.984	3
Class 3	0.799	0.745	0.966	4
Class 17	0.769	0.777	0.981	5
Class 4	0.749	0.685	0.939	6
Class 7	0.742	0.729	0.959	7
Class 2	0.703	0.65	0.941	8
Class 1	0.681	0.63	0.946	9
Class 8	0.68	0.681	0.995	10

* The DDI names and information are shown detail in the Appendix A.

**Table 4 cells-10-03092-t004:** XGBoost classifier performance on the validation dataset.

Type of Interaction *	Recall	Precision	F1	Number of Drug-Drug Pairs
15	0.68	0.75	0.65	7
6	0.71	0.73	0.60	165
3	0.73	0.57	0.64	570
17	0.46	0.45	0.38	10
4	0.65	0.67	0.59	670

* DDI type names and information are shown detail in the Appendix A.

**Table 5 cells-10-03092-t005:** HAINI performance compared with previous studies using chemical similarity.

	Recall	Precision	F1
Average performance *	0.734	0.783	0.758
Best performance **	0.921	0.778	0.838
Narjes Rohani et al.	0.899	0.373	0.527
Mei Liu et al.	0.493	0.434	N/A
Wen Zhang et al.	0.765	0.617	0.683

* Average performance of our model, ** the best performance of our model among all the DDIs.

## Data Availability

The datasets generated/analyzed for this study can be found in the GitHub repository https://github.com/tair-group/HAINI accessed on 8 November 2021.

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
