# Peer review of "Machine Learning-Based Prediction of Drug-Drug Interactions for Histamine Antagonist Using Hybrid Chemical Features"

_cells, 2021, doi:10.3390/cells10113092_

Round 1

Reviewer 1 Report

This study proposed a machine-learning model for identifying drug-drug interactions using chemical structures. A case study has been applied on histamine antagonist, which is an important pharmacovigilance assessment. The idea is of interest and the authors could reach promising performances on some key interactions. There are some points that need improvements before publication:

1.When comparing to previous works, did they also apply the model on histamine antagonist?

2.What are the differences between "experimental" and "investigational"? The authors should ensure that the testing did not appear in the training set.

3.Did the authors concern the drug similarity before entering to the models?

4.What are cut-off values for excluding the drug-drug interaction pairs?

5.The authors performed feature ranking for analysis, but they did not mention how many features that they used in the final model.

6.Where is the “Interaction Network Visualization”?

7.“Top 10 Random Forest metric performance values” (Fig. 2) did not make sense. The authors are suggested to revise this legend.

8.What is “chemical similarity”?

9.Equations should be assigned their numbers.

10. The references  for the evaluations metrics in line 267-270 should be provided. The authors could refer to  PMID: 33823302.

Reviewer 2 Report

This is an interesting paper, and has potential to be of pharmaceutical interest, though I have concerns about the veracity and impact of the SMILES-based methodology described. 

I recognized my concern, when I read the following sentence from the paper:

Lines 363-366:
However, if using only SMILE can be correctly predicted, it will greatly shorten the research process because SMILE is one of the factors that are easiest to achieve even in the initial stages of synthesizing a new drug.

This is a provocative statement to someone who understands what the SMILES string entails, as well as a bit about how they can be chopped up to make descriptors.

First of all, if your model guides you toward a certain combination of SMILES substrings that appear to give you desirable drug-interaction profiles, how do you back-propagate this to figure out how to reconnect those substrings to guide your synthesis priorities?

Secondly, one concern I have going through this paper is that it reports a reliance on SMILES as a determination of chemical structure but, unless one is relying explicitly on *canonical* SMILES, you do not have a unique one-to-one mapping.  I.e., for any given real chemical structure, there can actually be multiple regular (i.e., noncanonical) SMILES strings that fully describe that structure.  Thus, if one trains a model based on a single SMILES representation for every molecule, it will be possible to have pairs of molecules that share functionally analogous substructures, but differ completely in the descriptor set used to train the model.  Given this ambiguity, how can one be sure that one is is consistently perceiving the structural commonalities that truly dictate the drug interaction behavior?

The answer to me would be to impose a consistent canonicalization scheme on all structures used, or to instead rely on InChi as a structural representation, or (probably most commonly) to instead rely on graph theory.  There is a huge volume of graph theoretical chemoinformatics literature that pursues goals that are similar to those discussed here.  The graph-based models are particularly powerful, since they can be readily extended via kernels (and comparable) which can augment the pure connectivity information with context sensitivity that can mimic conformational dependence, electronic/electrostatic features, and more.

There are a fair number of published graph theoretical models for predicting drug-drug interactions (Google of PubMed search will reveal them).

I would also mention that there is a manuscript (currently under review in BMC Bioinformatics) that blends SMILES-based representation and graph theoretical encoding:  https://www.researchsquare.com/article/rs-598562/v1

Beyond that, there are a substantial number of minor typographical and grammatical errors, throughout, and the authors are a bit sloppy in adhering to the SMILES acronym, sometimes typing it as 'SMILE', and occasionally as 'SMILESS'.

Reviewer 3 Report

Given the short time for the review process I tried to read the paper, but  honestly I think that the manuscript is really hard to read and follow. For example, in “Materials and Methods” there are unneeded details about the models used (see for example subsection 2.3.1), while the essential information is missing. Indeed, it is unclear  to me which features the authors used, for example: “This equation gave us the most features (up to 3600 features)” at line 178 it is a bit obscure to me.  In the following just a short list of other issues:

There is an extra S in  SMILESS lines 93 and 95 (introduction), instead at line 227 an S i missed.In general SMILES is wrongly spelled throughout the manuscript.

The quality of the paper should be improved, for example: “therefore, we made a number of improvements in the model to improve the recall and precision values  evenly.”, line 286

Similarly: “At the same time, the ranking of the important features by the BestFirst method also presents higher feature values than the Random Forest method.” (line 300), it is really hard to read and understand. The authors state that they improved the quality of the models, but again it is unclear how they reached the goal. Maybe, they build the final models selecting only the most important features, they need to clarify his point.

A reference for the Synthetic Minority Oversampling Technique is missed .

“SMILE significantly improved the working performance of the HAINI machine learning Model” (line 349), what does it mean ? 
“Moreover, our prediction is based solely on the SMILE and thus can be applied at
early stage of drug development”, this sentence to me makes no much sense 

Finally, while the approach the authors used it is interesting, I believe the overall quality of the paper should be increased.

Round 2

Reviewer 2 Report

Most of my concerns have been at least partially addressed, so I think the manuscript can be published after minor further revision.

However, I still take issue with the following sentence (lines 379-380):

"However, if using only SMILES can be correctly predicted, it will greatly shorten the research process because SMILES is one of the factors that are easiest to achieve even in the initial stages of synthesizing a new drug."

Without the capacity to back-propagate the model toward explicit, unambiguous structural guidance, it is difficult to say how the method can, as implied, effectively guide the molecular design and optimization process.  The model may predict that some structures might lead to DDIs, and other structures may be less likely to lead to DDIs, but it doesn't seem to provide actionable insight along the lines of, for example, "In order to reduce the chance of DDIs, one should change substructure xxx into new substructure yyy."

Also, the manuscript does not acknowledge a large volume of graph theoretical research publications that accomplish very similar chemical informatics tasks using representations that, arguably, may be more flexible and informative than what is described herein.

Finally, there are a fair number of spurious hyphenations of non-hyphenated words.

Reviewer 3 Report

Given the short time for the reviewing process, honestly, it is quite hard to give a final impression about the manuscript. Nevertheless, while I believe the paper is interesting, personally I think the authors need to improve the overall quality of the English. Three are sentences like “the interaction factor was generated by multiplying as the equation:” that do not sound at all. In addition there are several inaccuracies, like:

  • Again sometimes SMILE is used instead SMILES (Simplified Molecular Input Line Entry System)
  • At page 5 they refer to drug A as: “DrugA”, or “Drug A”, or “drug A”. To use a well defined nomenclature is indeed important in terms of readability of the paper
  • Page 4, “fo-cused”, “de-scriptors” , “interac-tions”, why is there a “-” within those words?

They tried to improve materials and methods, but overall to me this section needs to be substantially rewritten. Instead of giving details about standard ML approaches, they should focus on how they build the features and the final training and test sets. 
